# Delivery of a Jagged1-PEG-MAL hydrogel with pediatric human bone cells regenerates critically sized craniofacial bone defects

Archana Kamalakar[1], Brendan Tobin[2,3], Sundus Kaimari[1,4], M Hope Robinson[1], Afra I Toma[1,4], Timothy Cha[1], Samir Chihab[5], Irica Moriarity[6], Surabhi Gautam[5], Pallavi Bhattaram[5,7], Shelly Abramowicz[1,8], Hicham Drissi[5,7,9], Andres Garcia[2,10], Levi Wood[2,4,10], Steven L Goudy[11]*

[1]Department of Pediatric Otolaryngology, Emory University, Atlanta, United States; [2]Parker H. Petit Institute for Bioengineering and Biosciences, Georgia Institute of Technology, Atlanta, United States; [3]School of Chemistry and Biomolecular Engineering, Georgia Tech College of Engineering, Atlanta, United States; [4]Wallace H. Coulter Department of Biomedical Engineering, Georgia Institute of Technology, Atlanta, United States; [5]Department of Orthopedics, Emory University, Atlanta, United States; [6]Neuroscience Program in College of Sciences, Georgia Institute of Technology, Atlanta, United States; [7]The Atlanta Veterans Affairs Medical Center Atlanta, Atlanta, United States; [8]Department of Surgery, Division of Oral and Maxillofacial Surgery, Emory University, Atlanta, United States; [9]Department of Cell Biology, Emory University, Atlanta, United States; [10]George W. Woodruff School of Mechanical Engineering, Georgia Tech College of Engineering, Atlanta, United States; [11]Department of Pediatric Otolaryngology, Children's Healthcare of Atlanta, Atlanta, United States

*For correspondence:
steven.goudy@emory.edu

Competing interest: The authors declare that no competing interests exist.

**Abstract** Current treatments for congenital and acquired craniofacial (CF) bone abnormalities are limited and costly. Conventional methods involve surgical correction, short-term stabilization, and long-term bone grafting, which may include problematic allografts and limited autografts. While bone morphogenetic protein 2 (BMP2) has been used for bone regeneration, it can cause bone overgrowth and life-threatening inflammation. Bone marrow-derived mesenchymal stem cell therapies, though promising, are not Food and Drug Administration approved and are resource intensive. Thus, there is a need for effective, affordable, and less side-effect-prone bone regenerative therapies. Previous research demonstrated that JAGGED1 induces osteoblast commitment in murine cranial neural crest cells through a NOTCH-dependent non-canonical pathway involving JAK2–STAT5. We hypothesize that delivery of JAGGED1 and induction of its downstream NOTCH non-canonical signaling in pediatric human osteoblasts constitutes an effective bone regenerative treatment. Delivering pediatric human bone-derived osteoblast-like cells to an in vivo murine bone loss model of a critically sized cranial defect, we identified that JAGGED1 promotes human pediatric osteoblast commitment and bone formation through p70 S6K phosphorylation. This approach highlights the potential of JAGGED1 and its downstream activators as innovative treatments for pediatric CF bone loss.

### eLife assessment

Therapeutic treatments for congenital and acquired craniofacial (CF) bone abnormalities are not well developed. This study provides **convincing** evidence for an innovative regenerative treatment for pediatric craniofacial bone loss using Jagged1-PEG-MAL hydrogel with pediatric human bone cells. The report is a **valuable** advance in this field.

## Introduction

Craniofacial (CF) injuries comprise more than 25% of injuries reported to the National Trauma Data Bank in the US every year (*Nardi et al., 2020*; *Wu et al., 2021*; *Choi et al., 2020*). Left untreated, CF injuries can severely impair critical daily functions related to breathing, speech, eating, and swallowing, thus requiring urgent repair (*Kumar et al., 2016*). Current methods to repair CF bone loss include the direct implantation of either allografts or autografts, and/or subsequent revision surgeries (*Grover et al., 2011*; *Roberts and Rosenbaum, 2012*). Although generally successful, these conventional treatments present several limitations. Bone donor sites include rib, fibula, iliac crest, scapula, distal tibia, and medial femoral condyle. All these donor sites have limited availability and are not anatomically like the bone they replace; osteotomies are required to shape them so that they resemble the shape of the CF bones (*Kakabadze et al., 2017*; *Chim et al., 2010*). The allograft survival rate following an iliac graft procedure is 96.1% and the risk of infection ranges from 5% to 33% (*Roberts and Rosenbaum, 2012*; *Chaushu et al., 2010*). For maxillary or mandibular bone replacement, the iliac crest is the preferred graft source due to its robust corticocancellous anatomy. Risks associated with iliac crest grafting include significant pain at the donor site, nerve injury, decreased load bearing on the ipsilateral leg, and increased risk of hip fractures (*Dahlin and Johansson, 2011*; *Baumhauer et al., 2014*). Due to the limited supply of bone, revision bone graft surgeries are often required and are expensive ($35,000–$52,000 per patient) and cause great discomfort (*Laurie et al., 1984*). These and other complications undermine the patient's quality of life in addition to the social stigma resulting from the facial deformity, which can also lead to psychological distress (*De Sousa, 2010*).

CF bone development primarily occurs through intramembranous ossification, a process where pre-osteoblasts mineralize directly, without a cartilage intermediate, making it distinct from long bone development (endochondral ossification) (*Shah et al., 2021*). Intramembranous ossification recruits cranial neural crest (CNC) cells as osteoblast precursors during CF bone development (*Shah et al., 2021*). Extensive efforts have been made toward using parathyroid hormone (PTH (1-34)), vascular endothelial growth factor (VEGF), fibroblast growth factor (FGF), stromal cell-derived factor-1, or transforming growth factor-beta 2 (TGFβ2) as alternatives to bone regenerative treatments in preclinical models but resulted in limited success as an in vivo treatment option (*Rowshan et al., 2010*; *Elsalanty and Genecov, 2009*; *Nosho et al., 2020*; *Takayama et al., 2017*; *Albanese et al., 2013*). Platelet-rich plasma supplemented with various biological growth factors such as VEGF, FGF, and TGFβ2 have also been used to enhance wound healing, chemotaxis, angiogenesis, proliferation of mesenchymal stem cells, and osteoblasts. Many of these studies demonstrated potential improvement in bone healing; however, these strategies face significant translational barriers (*Albanese et al., 2013*). Bone morphogenetic protein-2 (BMP2) is a Food and Drug Administration (FDA)-approved bone regenerative strategy along with the delivery of stem cells. BMP2 is used to reconstruct spine and maxillary bone in adults (*Seto et al., 2006*; *Zuk, 2008*). BMP2 treatment can lead to ectopic bone growth, hypertrophy and life-threatening side effects (e.g., uncontrolled inflammation), which may accelerate bone loss (*Huang et al., 2014*; *Tannoury and An, 2014*). The use of BMP2 for the treatment of pediatric cases of CF trauma is not FDA approved due to concerns of severe swelling of the face and airways. Although generally considered safer, stem cell-based treatments are time consuming, have heterogenous results and add to the high expense of repairing CF bone loss (*Zuk, 2008*). Thus, there is a critical need for novel bone regenerative therapies to treat CF bone loss that have minimal side effects, are readily accessible, and affordable.

The NOTCH signaling pathway is involved in many cellular processes, including determination of cell fate, and has been explored as a potential target for regeneration of long bone injuries (*Zanotti and Canalis, 2012*). NOTCH signaling occurs via cell-to-cell binding of a NOTCH ligand (e.g., JAG1) to a NOTCH receptor. Internalization of the NOTCH intracellular domain leads to the expression of canonical NOTCH genes *HES1* and *HEY1*, which are known to have both osteo-inductive and

osteo-inhibitory roles, thus obfuscating the effectiveness of NOTCH-based bone regenerative therapies (*Nobta et al., 2005*; *Regan and Long, 2013*). However, we and others have demonstrated that JAG1 exhibits osteo-inductive properties, as demonstrated by induction of pre-osteoblast genes like *Runx2* in murine CNC cells, even when the canonical NOTCH pathway is inhibited (*Kamalakar et al., 2019*; *Youngstrom et al., 2017*). Our recent publications further identified a JAG1–JAK2 non-canonical signaling pathway that promotes murine CNC commitment to osteoblast differentiation in mice (*Kamalakar et al., 2019*).

This unexpected finding raises critical questions about the role of non-canonical JAG1 signaling during human CF regeneration. In this study, we postulate that (1) JAG1 can induce osteoblast differentiation and mineralization of pediatric human bone-derived osteoblast-like (HBO) cells, and (2) the delivery of JAG1 non-canonical signaling constitutes an effective treatment for inducing bone regeneration in a pediatric, preclinical CF bone loss model. Consequently, we evaluated (1) the ability of JAG1 to regenerate bone in a pediatric critically sized CF defect murine model when delivered in a synthetic hydrogel coupled with pediatric HBO cells, and (2) characterized the downstream JAGGED1 non-canonical signaling mechanisms. Our results reveal potential treatment options in the form of JAG1 and/or its downstream targets to induce bone regeneration in CF bone loss injuries and provide alternative treatment options for CF defects.

## Results

### JAG1 induces mineralization of pediatric human bone-derived osteoblast-like cells

To test whether JAG1 induces osteoblast commitment and differentiation in human osteoblast-like primary cells, we derived human bone osteoblast-like (HBO) cell lines (HBO1, 2, 3, 4, 5, 6, and 7) from seven healthy pediatric human bone samples, as described (*Lockyer et al., 2007*; *Figure 1—figure supplement 1*). HBO cells were then treated with Fc-Dynabeads (Fc-bds) (5.7 µM) as a negative control, since JAG1 is a chimeric recombinant protein with an Fc-portion, and JAG1-Dynabeads (JAG1-bds) (5.7 µM), in the presence of osteogenic media. On day 21, the cells were stained for mineralization using Alizarin Red S stain (*Figure 1A, B*). We observed significantly increased mineralization in JAG1-bds-treated samples compared to growth media-, osteogenic media-, and Fc-bds-treated (p = 0.0107) cells. Additionally, PCR analysis of HBO1 cells from a repeat experiment collected at days 7, 14, and 21 showed significantly increased expression of osteogenic genes with JAG1-bds stimulation (*Figure 1C*). *ALPL* was significantly expressed at day 7, with a 3.5-fold increase (p = 0.0004) compared to HBO1 cells grown in growth media. In contrast, significant expression levels of *COL1A1* and *BGLAP* were observed at 14 days, with a 5.1-fold increase (p = 0.0021) of *COL1A1* and a 12.3-fold increase (0.0002) of *BGLAP* when compared to growth media conditions. Interestingly, while some mineralization is observed in the osteogenic media and Fc-bds (*Figure 1A*) conditions, there were no significant increases in osteogenic gene expression (*Figure 1C*). Expression of *RUNX2* and *SP7* was not significantly altered across all conditions and time points (not shown). In preparation for in vivo studies, HBO cells were incorporated into JAG1-Dynabead-PEG-4MAL and grown in vitro followed by Alizarin Red S staining to assess mineralization (*Figure 1—figure supplement 2*). These data indicate that JAG1 can induce osteoblast commitment, differentiation, and mineralization of pediatric HBO cells.

### Delivery of JAG1-Dynabead-PEG-4MAL with pediatric HBO cells repairs critically sized cranial defects

Since we observed an induction of osteoblast commitment and differentiation in pediatric HBO cells in vitro, we next assessed whether co-delivery of JAG1-presenting hydrogels with pediatric HBO cells act similarly in murine CF defects. We recently reported that JAG1 can induce murine CNC cell osteoblast commitment and repair cranial bone defects in vivo (*Kamalakar et al., 2021*). To establish a more translatable use of JAG1 in treating CF bone loss, we assessed whether JAG1 can stimulate human cells (pediatric HBO) to facilitate bone regeneration in NOD-SCID mice, to prevent graft rejection of the HBO cells. JAG1-Dynabead-PEG-4MAL hydrogels also encapsulating pediatric HBO cells obtained from three separate donors were implanted in critically sized parietal bone defects (4 mm) in NOD-SCID mice (n = 4–6 per donor, 13–15 total) (*Figure 2—figure supplement 1*). The volume of bone regenerated by the JAG1-PEG-4MAL-pediatric HBO hydrogel with or without DAPT,

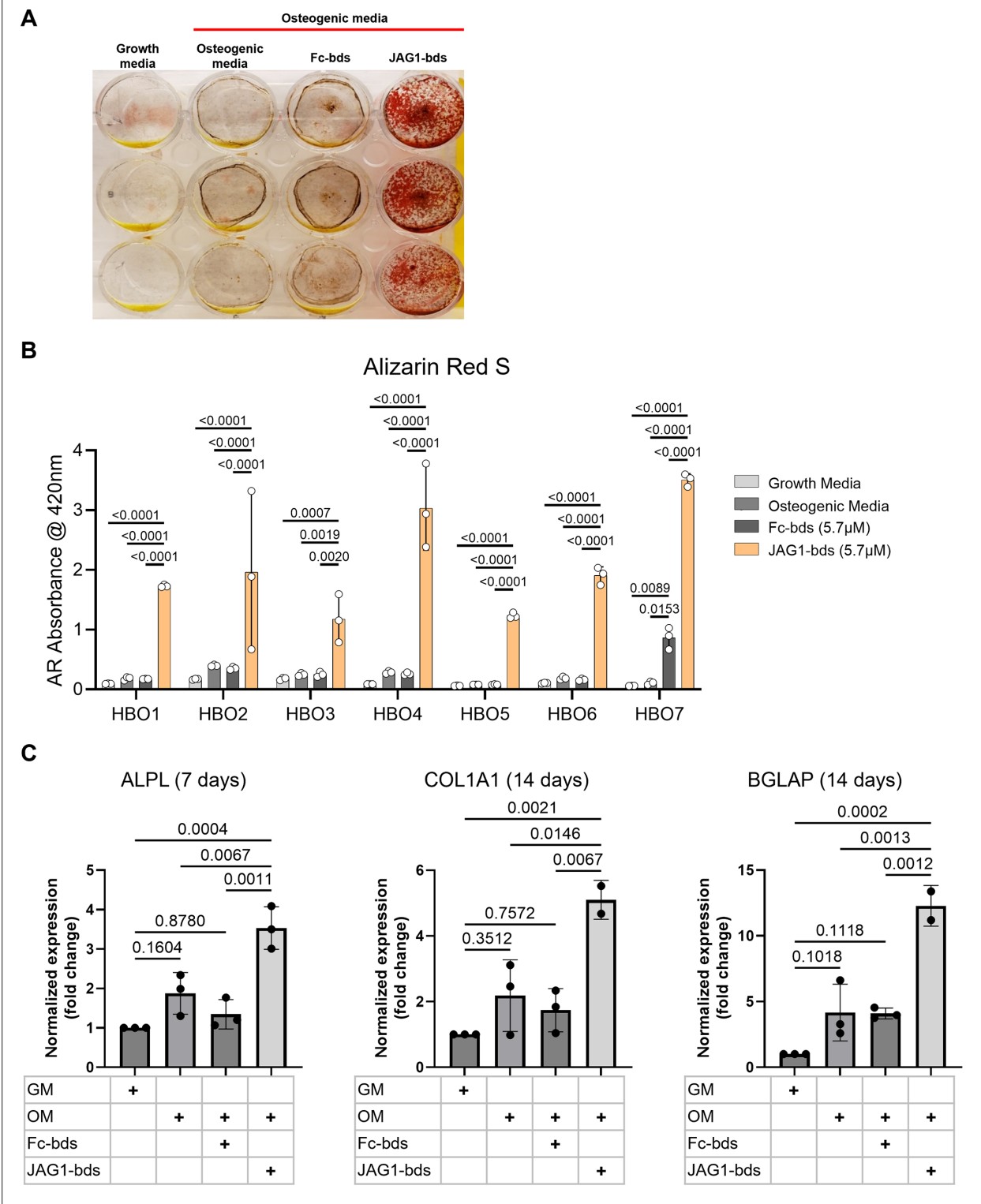

**Figure 1.** JAGGED1-induced mineralization and gene expression in HBO cells: seven HBO cell lines were treated with growth media alone, osteogenic media alone or with Fc-Dynabeads (5.7 µM) or JAG1-Dynabeads (5.7 µM). The cells were half-fed every 5 days. On day 21 cells were fixed with 50% ethanol and thereafter, stained with 1% Alizarin Red S. (**A**) Representative image of HBO2. (**B**) Alizarin Red S dye was extracted from Alizarin Red S-stained cells using a 1:10 dilution of acetic acid and water, and the absorbance was read at 420 nm. Data represent the mean values of three technical replicates per cell line (mean ± standard deviation [SD], one-way analysis of variance [ANOVA] with Tukey post hoc). (**C**) HBO1 primary cell line was grown in triplicate and treated with growth media alone, osteogenic media alone or with Fc-Dynabeads (5.7 µM) or JAG1-Dynabeads (5.7 µM). The

*Figure 1 continued on next page*

Figure 1 continued

cells were half-fed every 5 days and collected at 7, 14, and 21 days. qRT-PCR was performed (see Methods). Data were normalized to growth media with glyceraldehyde-3-phosphate dehydrogenase (GAPDH) as the reference gene. Data represent the mean values of three biological and two technical replicates per condition (mean ± SD, ordinary one-way ANOVA with Šídák's multiple comparisons test, with single pooled variance).

The online version of this article includes the following figure supplement(s) for figure 1:

**Figure supplement 1.** Method of processing human bone samples to produce primary human bone-derived osteoblast-like cell lines.

**Figure supplement 2.** PEG-4MAL hydrogel-encapsulated JAGGED1-induced mineralization of HBO cells.

an inhibitor of NOTCH canonical signaling, was measured using micro computed tomography (μCT), and compared to HBO cells alone, Fc-bds (20 μM) + BMP2 (2.5 μM) treatments. To maintain continuous osteogenic induction, additional JAG1-PEG-4MAL-pediatric HBO cells and control hydrogels were injected into the defects transcutaneously at week 4. After 8 weeks, we quantified differences in bone volume (BV) within the cranial defect using μCT analysis (*Figure 2A, B*). We observed that there was minimal bone regenerated in mice treated with cells alone. As expected, BMP2 significantly increased regenerated BV (p = 0.0002) compared to the cells alone group. The BV regenerated by JAG1-bds in the absence (fold change: 1.5) and presence of DAPT (fold change:1.6) was significantly higher compared to the cells alone treatment group (p = 0.0092 and 0.0021, respectively). There was no sex-based difference in regenerated BV. An initial pilot study also demonstrated no difference in bone regeneration between an Empty Defect model (no HBO cells) and a Cells Alone group (*Figure 2—figure supplement 2*). In *Figure 2C*, paraffin sections of mouse skulls were stained with Masson trichrome stain, as described in methods under histology. Qualitative assessments of the stained sections revealed increased collagen (blue color) in samples obtained from mice treated with BMP2, as expected, and with JAG1-bds and JAG1-bds + DAPT compared to the mice treated with cells alone. To validate the results observed in the samples stained with Masson trichrome, we performed immunohistochemical 3,3'-diaminobenzidine (DAB) staining with additional sections from the same mice using rabbit IgG monoclonal anti-COL1A1 antibody. These results corroborate what was revealed with the Masson trichrome staining, that mice treated with cells alone had less collagen production (*Figure 2—figure supplement 3*). This suggests that JAG1 can be used as a bone regenerative therapy where JAG1 induces bone regeneration independently of NOTCH canonical signaling in human cells.

## Transcriptional profiling of cultured HBO cells reveals genes regulated by the non-canonical NOTCH pathway

We previously found that JAG1 can activate a NOTCH non-canonical JAK2–STAT5 signaling pathway in mouse CNC cells, which stimulated expression of osteoblast genes (*Runx2* and *Bglap* (Osteocalcin)) as well as osteoblast commitment and proliferation (*Kamalakar et al., 2019*; *Kamalakar et al., 2021*). Thus, we asked if JAG1 would have similar effects for a non-canonical NOTCH pathway on HBO cells. To evaluate this, we cultured HBO cells from a single donor in triplicate and conditioned for 24 hr with vehicle, DAPT, JAG1-bds, or both. Comparison of the JAG1-bds and JAG1-bds + DAPT conditions revealed clusters of genes that were up- or downregulated by JAG1 stimulation and remained with inhibition of NOTCH, that is, defining the non-canonical pathway (*Figure 3A*). Analysis of differentially expressed genes (DEGs) in JAG1-bds and JAG1-bds + DAPT groups compared to no treatment revealed a total of 448 upregulated genes and 435 downregulated genes in the non-canonical pathway (*Figure 3B*, *Supplementary file 1*). These include upregulation of genes involved in osteoblast commitment (*RUNX2*), matrix remodeling (*MMP3*), and diverse cytokines and chemokines (*CCL5*, *CXCL1*, and *CXCL6*) as part of the non-canonical pathway (*Figure 3C*). Gene ontology (GO) analysis of the upregulated genes in the non-canonical pathway revealed significant over-representation of GO terms associated with *RUNX2*, cytokine signaling, Nuclear factor kappa B (NF-κβ), and cell cycle (*Figure 3D*). More interestingly, the PIP3 activating *AKT* signaling pathway was upregulated by JAG1-bds treatment, suggesting that JAG1 can activate NOTCH non-canonical signals via the AKT pathway.

Collectively, these data suggest that JAG1 has a profound NOTCH non-canonical effect on HBO cells that stimulates HBO cell-osteoblast commitment and differentiation leading to HBO-cell-induced bone formation.

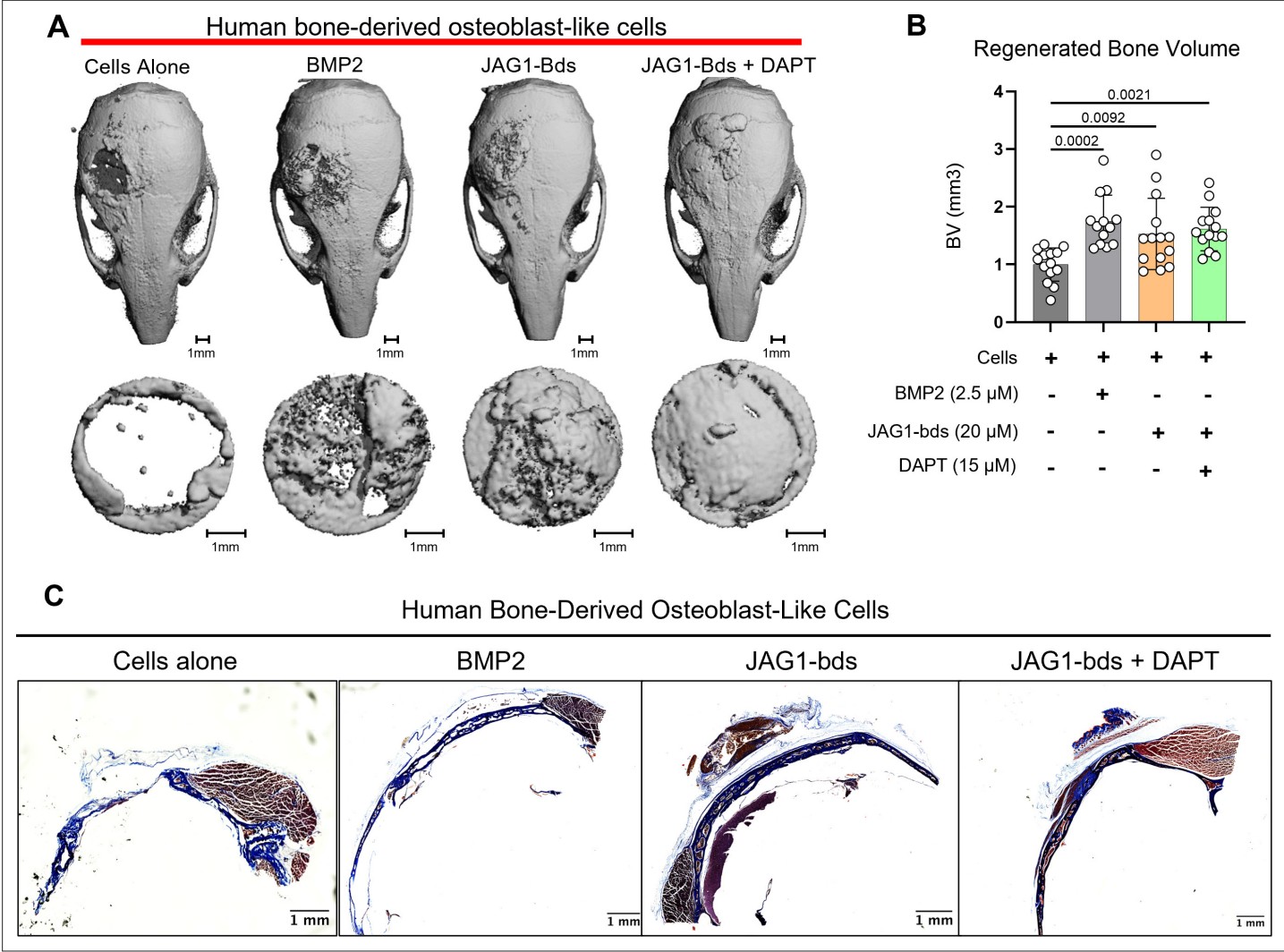

**Figure 2.** JAG1 delivery in a PEG hydrogel stimulates bone regeneration in a critical-sized bone defect mouse model. HBO cells alone or in the presence of JAG1-Dynabeads complex (20 µM) ± DAPT and bone morphogenetic protein 2 (BMP2; 2.5 µM) + Fc-Dynabeads were incorporated in 4% PEG-MAL hydrogels and implanted into 4 mm critical-sized defects in the parietal bones of 6- to 8-week-old NOD SCID mice ($n = 4–6$ per HBO cell donor, 13–15 total) as two separate doses (Initial dose, week 4). After 8 weeks, we quantified differences in regenerated bone volume within the defect and compared them between experimental groups by micro computed tomography (µCT) analysis. (**A**) µCT reconstructions of defects. (**B**) Quantification of regenerated bone volume. Data are presented as mean ($n = 13–15$) ± standard deviation (SD) with p-values reported (one-way analysis of variance [ANOVA] with Šídák's multiple comparisons test). (**C**) shows representative sections of the defect area on skulls from mice from all experimental groups stained with Masson trichrome stain.

The online version of this article includes the following figure supplement(s) for figure 2:

**Figure supplement 1.** Visual depiction of calvarial defect studies.

**Figure supplement 2.** Pilot study of JAG1-bds delivery in a PEG hydrogel stimulates bone regeneration in a critical-sized bone defect mouse model.

**Figure supplement 3.** Immunohistochemical staining of calvarial defect tissue for Collagen 1.

## JAG1 induces increased cytokine production and phosphorylation of NOTCH non-canonical pathway targets in pediatric HBO cells

Having observed that JAG1-induced osteoblast commitment of murine CNC cells via a NOTCH non-canonical pathway (JAG1–JAK2), we sought to determine whether JAG1 activated NOTCH non-canonical pathways and targets in the pediatric HBO cells. Serum-starved pediatric HBO cells were subsequently treated with JAG1-bds (5.7 µM) with or without DAPT (15 µM), to block the NOTCH canonical pathway, as a time course stimulation for 5, 10, 15, and 30 min. Phosphorylation levels for signaling molecules were assessed via Luminex-based multiplex assays. We observed significantly

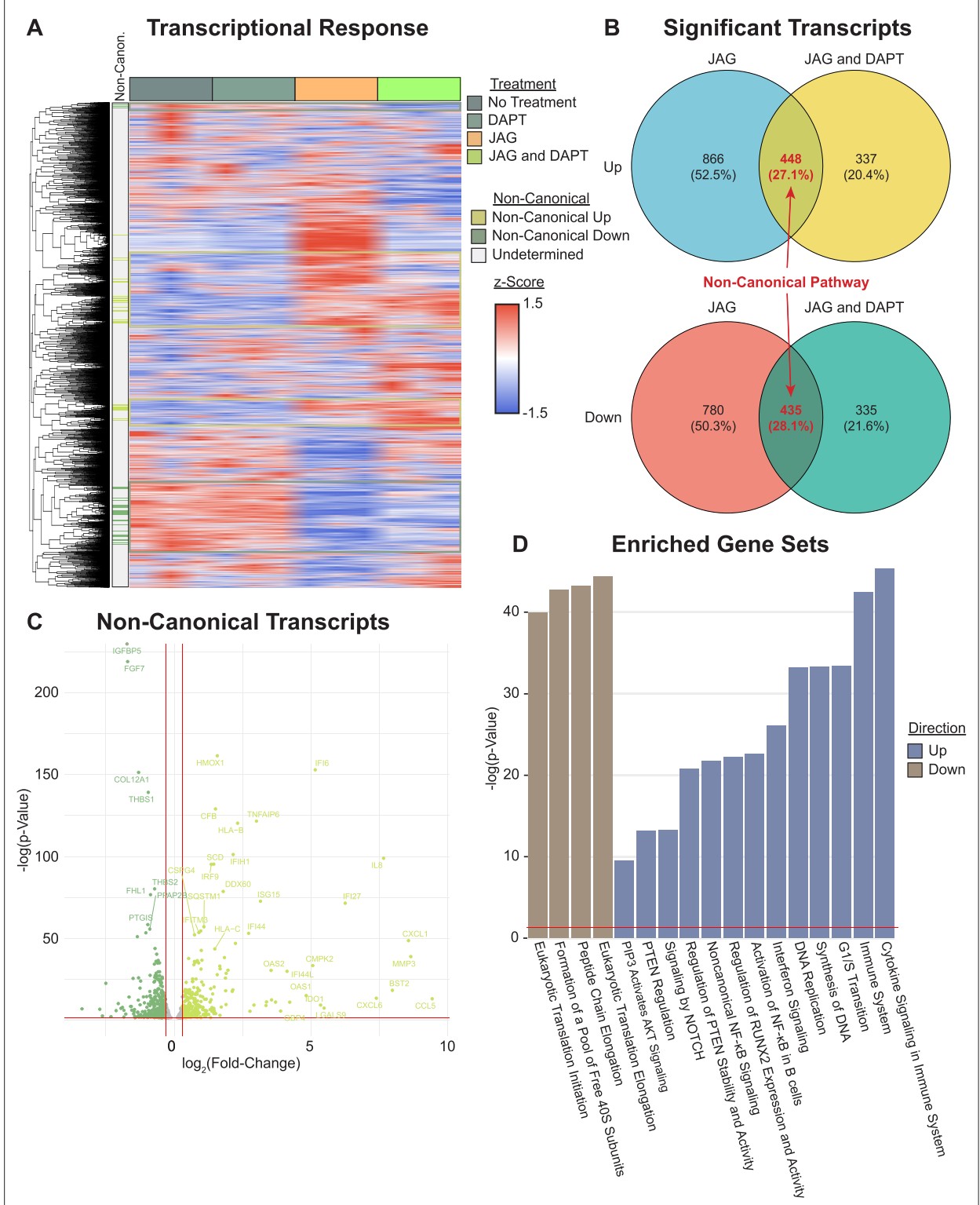

**Figure 3.** Transcriptional profiling reveals genes and pathways stimulated by non-canonical NOTCH signaling. (**A**) RNAseq reveals clusters of genes associated with the non-canonical NOTCH pathway (rows are z-scored, side color bar identifies up- and downregulated differentially expressed genes (DEGs) stimulated by the non-canonical pathway). (**B**) Overlapping DEGs in the JAG1-bds vs no-treatment and JAG1-bds + DAPT vs no-treatment comparisons reveal the non-canonical pathway (DEseq2). (**C**) Overlapping DEGs from JAG1-bds + DAPT vs no-treatment comparison. (**D**) Gene ontology over-representation test reveals significantly enriched up- and downregulated pathways (false discovery rate adjusted).

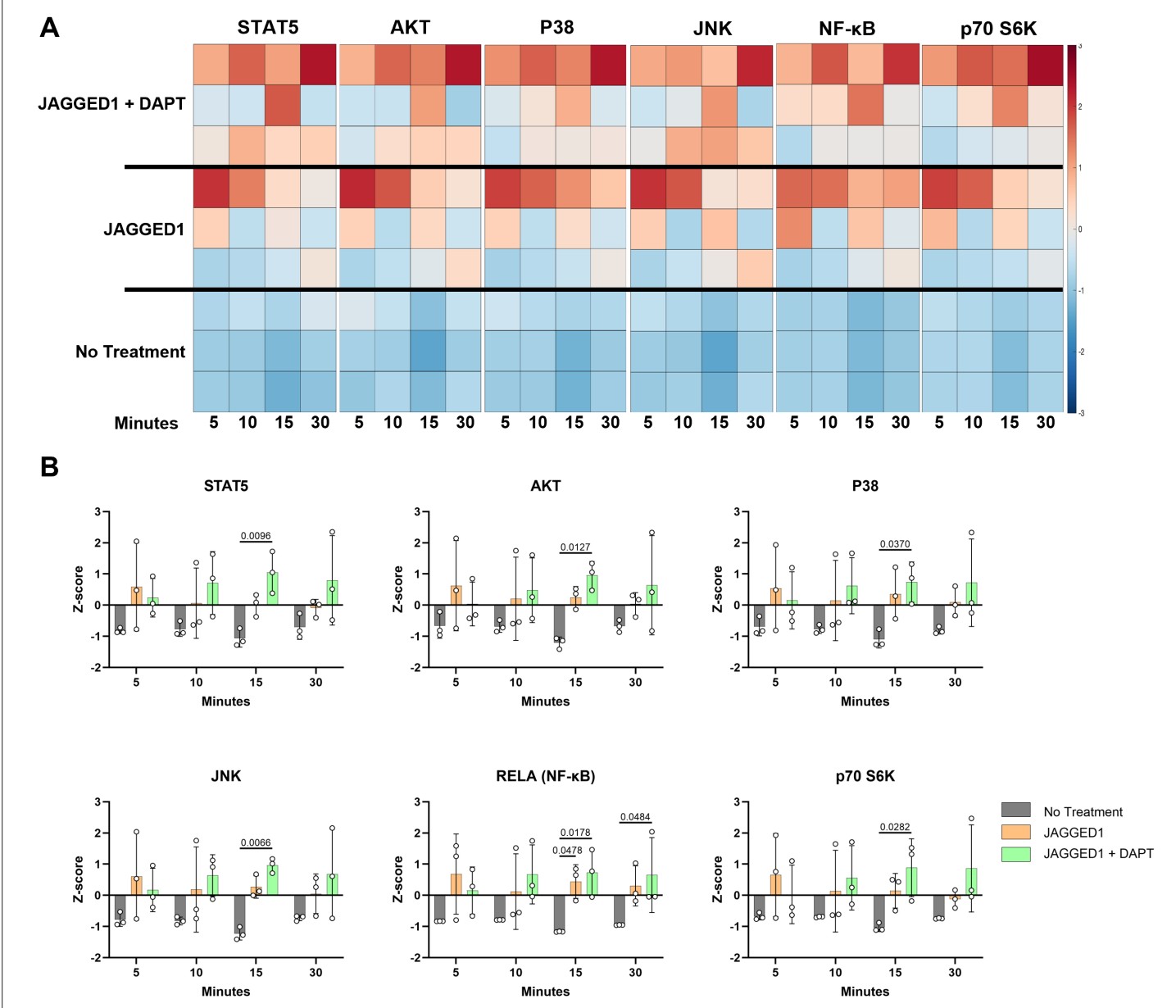

**Figure 4.** JAGGED1 induces a non-canonical NOTCH pathway in HBO cells. HBO cells undergo mineralization through a non-canonical pathway. Luminex analysis of lysates obtained from three HBO cell lines untreated or treated with Dynabead-bound recombinant JAG1-Fc fragment (5.7 µM) ± DAPT (15 µM), a NOTCH canonical pathway inhibitor in a time course manner (5, 10, 15, and 30 min), (**A**) Heatmaps and (**B**) z-Scores plotted on graphs. Each data point represents mean *n* = 3 ± standard deviation (SD) per cell line with p-values reported.

increased phosphorylation of multiple signaling molecules, including STAT5, AKT, P38, JNK, RELA, and p70 S6K in JAG1-bds-treated cells, even in the presence of DAPT (*Figure 4*). We have previously shown that JAG1 induced the phosphorylation of STAT5 during CNC cell differentiation to osteoblasts (*Kamalakar et al., 2021*). Prior studies emphasize the importance of non-canonical signaling, crosstalk between NOTCH, and other cellular signaling mechanisms. For example, the WNT pathway cross-talks with the NOTCH canonical pathway during vascular morphogenesis (*Caliceti et al., 2014*). Furthermore, previous reports have shown that STAT5 is essential for AKT–p70 S6K activity during lymphocyte proliferation in patients with leukemias and lymphomas (*Lockyer et al., 2007*). Similarly, the P38 pathway has been shown to activate the mammalian target of rapamycin (mTOR)–p70 S6K pathway during oxidative stress in mouse embryonic fibroblast cells which culminates in upregulation of antioxidative enzymes that assist in reactive oxygen species removal and thereafter increase cell

survival (*Gutiérrez-Uzquiza et al., 2012*). Also shown previously, JNK phosphorylates p70 S6K to induce osteoblast proliferation and differentiation of MC3T3 cells, and can crosstalk in other physiological systems, for example, during hepatocyte proliferation (*Bouxsein et al., 2010*; *Love et al., 2014*; *Zhao et al., 2014*). A study by Miwa et al., in 2012, showed that AKT–mTOR–p70 S6K, extracellular signal-regulated kinases (ERK), and NF-κB were involved together in proliferation of osteosarcoma cells and these pathways could be inhibited by caffeine thereby decreasing tumor burden (*Miwa et al., 2012*). We observe in our results that many of the pathways activated by JAG1 in the pediatric HBO cells lead to the phosphorylation of p70 S6K, downstream. These findings implicating phospho-protein signaling pathways, especially NF-κB and AKT (downstream of phosphatase and tensin homolog (PTEN)) are consistent with our transcriptional profiling (*Figure 3*) and collectively implicate p70 S6K as a potential downstream contributor to JAG1-induced, pediatric HBO cell-mediated bone regeneration. Therefore, we proceeded to determine if p70 S6K is an essential downstream target of the JAG1-induced NOTCH non-canonical signaling in pediatric HBO cells (*Figure 5*).

## Inhibition of phosphorylated p70 S6K leads to inhibition of JAG1-induced mineralization of pediatric HBO cells

As shown previously, we measured significantly increased alkaline phosphatase production as well as mineralization in JAG1-bds-treated samples compared to all other treatments. We also showed an increase in the phosphorylation of p70 S6K. According to multiple reports in literature, phosphorylation of p70 S6K occurs downstream of multiple pathways (AKT, JAK–STAT, and P38) that we identified in JAG1-induced pediatric HBO cells (*Lockyer et al., 2007*; *Gutiérrez-Uzquiza et al., 2012*; *Miwa et al., 2012*). Thus, we next tested whether the phosphorylation of p70 S6K is an essential downstream target of JAG1-NOTCH during HBO cell mineralization. HBO cells were treated with growth media alone, osteogenic media alone, or Fc-bds (5.7 µM) as negative controls, and JAG1-bds (5.7 µM) with or without S6K-18, an inhibitor of phosphorylated p70 S6K (*Ye et al., 2011*), in the presence of osteogenic media. On day 21, the cells were stained for mineralization using Alizarin Red S stain. As shown in *Figure 5*, we observed that negative controls did not induce mineralization of the HBO cells while JAG1-bds induced significantly higher levels of mineralization, as expected. S6K-18-treated samples showed partial inhibition of 50.2% of the JAG1-bds-induced mineralization (p = 0.0015) and not 100%, possibly because the NOTCH canonical pathway was not inhibited in these samples. Inhibition of JAG1-bds-induced mineralization by S6K-18 treatment alone was significant compared to mineralization induced by JAG1-bds alone suggesting that the phosphorylation of p70 S6K is an essential event downstream of JAG1-NOTCH in JAG1-stimulated HBO cells. Additionally, inhibition of JAG1-bds-induced mineralization was observed with DAPT, an inhibitor of canonical NOTCH signaling (*Figure 5—figure supplement 1*). Subsequent repeat experiments involved collecting the cells on days 9, 14, and 21 for qRT-PCR analysis. While inhibition of NOTCH and p70 S6K decreased mineralization in our mineralization assay, there are no statistically significant changes in gene expression for *ALPL*, *COL1A1*, or *BGLAP* (*Figure 5—figure supplement 2*). These results suggest that the HBO cells phenotypes are maturing into osteocytes and that inhibiting p70 S6K hinders the cellular ability to mineralize but not the cell phenotype progression.

## Discussion

We and others have previously demonstrated that JAG1 exhibits osteo-inductive properties in murine cell lines (*Kamalakar et al., 2019*; *Kamalakar et al., 2021*; *Kornsuthisopon et al., 2022*; *Youngstrom et al., 2016*; *Dishowitz et al., 2014*). JAG1 also induces the expression of pre-osteoblast genes like *Runx2* in murine CNC cells, even when the canonical NOTCH pathway is disabled using DAPT (*Kamalakar et al., 2019*; *Youngstrom et al., 2017*), and further promotes CNC cell commitment to osteoblast differentiation via a JAG1–JAK2 non-canonical signaling pathway (*Kamalakar et al., 2019*; *Youngstrom et al., 2016*). Thus, we hypothesized that JAG1 can induce osteoblast differentiation and mineralization of pediatric human bone osteoblast-like (HBO) cells, and the delivery of JAG1 with pediatric HBO cells constitutes an effective treatment for inducing bone regeneration in a pediatric CF bone loss model.

To enhance the clinical translatability of this study, our lab derived HBO-like cells by collagenase digestion of human pediatric fibula bones. JAG1 has previously been shown to induce survival and

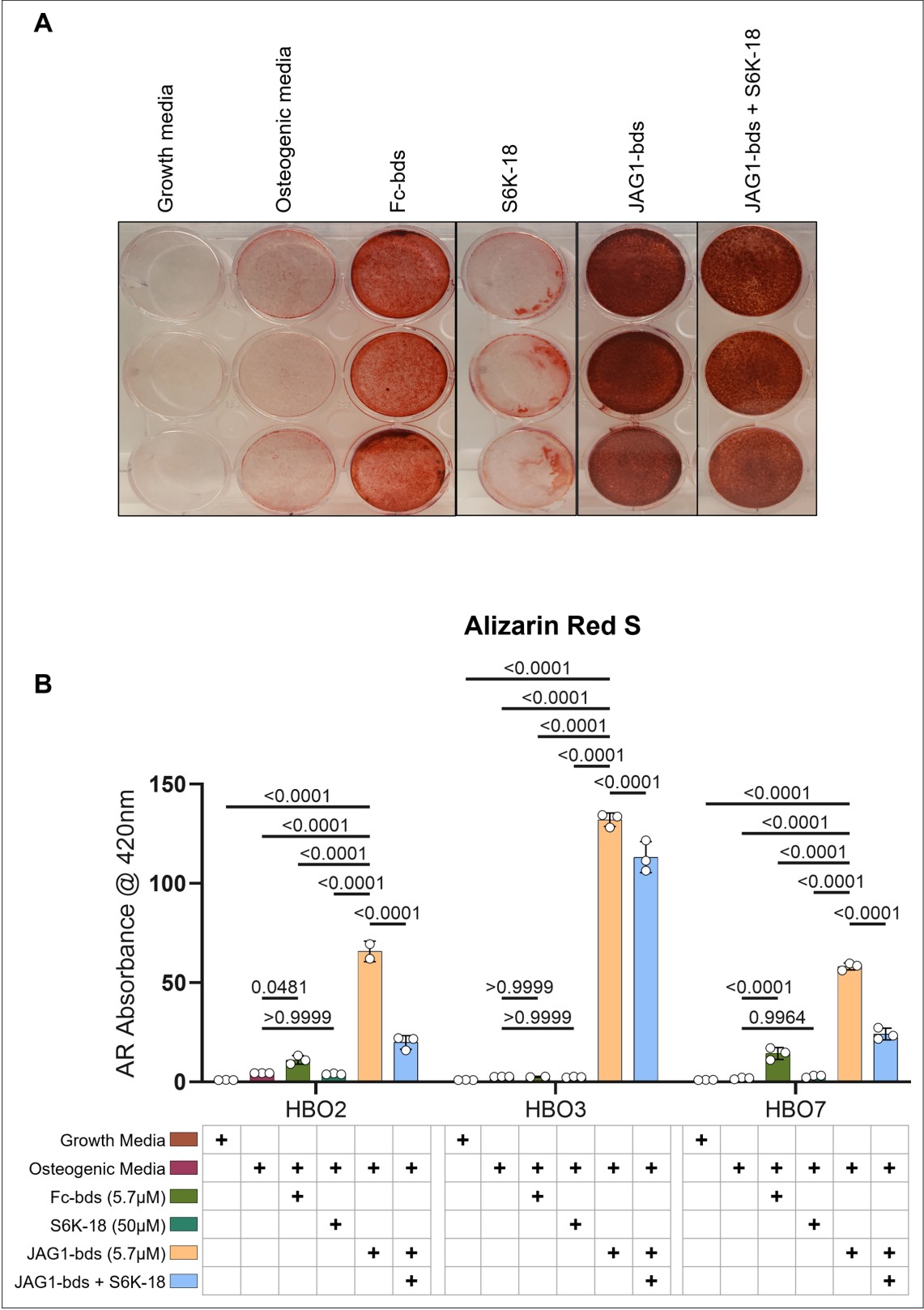

**Figure 5.** p70 S6K is an essential target during JAGGED1-induced mineralization of HBO cells: HBO cells were treated with growth media alone, osteogenic media alone or with Fc-Dynabeads (5.7 μM), S6K-18 alone (a p70 S6K phosphorylation inhibitor) (50 μM), and JAG1-Dynabeads (5.7 μM) alone or in combination with S6K-18 (50 μM). The cells were half-fed every 5 days. On day 21 cells are fixed with 50% ethanol and thereafter, stained

*Figure 5 continued on next page*

*Figure 5 continued*

with 1% Alizarin Red S.(**A**) Representative image of HBO7. (**B**) Alizarin Red S dye was extracted from stained cells using a 1:10 dilution of acetic acid and water, and the absorbance was read at 420 nm. Data represent mean *n* = 3 ± standard deviation (SD) per cell line with p-values indicated.

The online version of this article includes the following figure supplement(s) for figure 5:

**Figure supplement 1.** Mineralization assay with DAPT inhibition of the NOTCH canonical pathway.

**Figure supplement 2.** Inhibition of JAGGED1-induced mineralization of HBO cells with inhibitors of NOTCH and p70 S6K.

proliferation of human mesenchymal stem cells, which are osteoblast precursors and various other human cell types, for example LNCaP which is a prostate cancer cell line, glioma cells, and intestinal epithelial cells during progression of colorectal cancer (*Pannequin et al., 2009*; *Purow et al., 2005*). Osathanon et al. showed that tissue culture plate surface-immobilized JAG1-stimulated osteoblast proliferation and differentiation in iliac bone-derived cells (*Osathanon et al., 2019*). As seen in *Figure 1A, B*, we also observed increased mineralization of the JAG1-bds-treated HBO cells compared to other controls (growth media alone, osteogenic media alone, and Fc-bds) in HBO cell lines obtained from seven different pediatric fibular samples. The ability of Fc-bds to stimulate mineralization in vitro is previously described in the literature but has not been found to be osteogenic in vivo (*Kamalakar et al., 2019*; *Kamalakar et al., 2021*). This suggests that JAG1 reliably induces the human osteoblast-like primary cell expansion and differentiation in multiple human bone cell lines. These results suggest that HBO cells behave similar to murine CNC cells in vitro and applying these humanized experiments to critically sized defects will assess JAG1 as a potential bone regenerative therapeutic.

To confirm that JAG1 induces pediatric HBO cells to facilitate osteogenesis in vivo, as it did in vitro, we tested this strategy using an in vivo model of repair using murine CF defects. As shown in *Figure 2—figure supplement 1A, B*, PEG-4MAL hydrogels encapsulating JAG1-Dynabeads and pediatric HBO cells were implanted in critically sized parietal bone defects (4 mm) in athymic nude mice (NOD-SCID), which were used to prevent rejection of human cells. Four weeks later the mice received a second dose of treatments (*Figure 2—figure supplement 1C*) via a transcutaneous injection to maintain the osteogenic signaling. Intermittent treatment with bone-regenerative therapeutics, like PTH (*Kamalakar et al., 2019*; *Nardi et al., 2020*; *Wu et al., 2021*; *Choi et al., 2020*; *Kumar et al., 2016*; *Grover et al., 2011*; *Roberts and Rosenbaum, 2012*; *Kakabadze et al., 2017*; *Chim et al., 2010*; *Chaushu et al., 2010*; *Dahlin and Johansson, 2011*; *Baumhauer et al., 2014*; *Laurie et al., 1984*; *De Sousa, 2010*; *Shah et al., 2021*; *Rowshan et al., 2010*; *Elsalanty and Genecov, 2009*; *Nosho et al., 2020*; *Takayama et al., 2017*; *Albanese et al., 2013*; *Seto et al., 2006*; *Zuk, 2008*; *Huang et al., 2014*; *Tannoury and An, 2014*; *Zanotti and Canalis, 2012*; *Nobta et al., 2005*; *Regan and Long, 2013*; *Youngstrom et al., 2017*; *Phelps et al., 2012*; *Bouxsein et al., 2010*; *Love et al., 2014*; *Zhao et al., 2014*; *Sayers et al., 2021*; *Thomas et al., 2022*; *Fowler et al., 2012*; *Langdahl et al., 2018*) has been shown to lead to anabolic increase in bone mineral density. The ORTHOUNION clinical trial which aimed at enhancing bone healing in long bone nonunion fractures focused on testing the treatment involving two different sequential doses of expanded bone marrow-derived mesenchymal stem cells (*Gómez-Barrena et al., 2018*). In our study, 8 weeks after the initial dose, the regenerated BV was measured using µCT. As seen in *Figure 2B*, the fold change of BV regenerated by JAG1-bds in the presence of DAPT (fold change: 1.6, normalized to cells alone treatment) was comparable to JAG1-bds alone treatment (fold change: 1.5, p = 0.9583), and was significantly higher compared to the cells alone treatment group (p = 0.0038). The inhibition of NOTCH canonical signaling in mesenchymal stem cells using DAPT has been shown previously to enhance osteogenesis (*Luo et al., 2019*). A rheumatoid Arthritis C57BL/6 SCID mouse model carrying a human *TNF* transgene when treated with intermittent doses of DAPT showed improved bone regeneration (*Zhang et al., 2014*). This supports our current results that JAG1 induces bone regeneration independently of NOTCH canonical signaling in the HBO cells. However, structural and mechanical properties of the bone will require characterization in the future using biomechanical testing.

As shown in our previous publications, JAG1-induced osteoblast commitment of murine CNC cells via a NOTCH non-canonical pathway. In our current data, it is observed that JAG1-induced HBO cells to regenerate bone and repair cranial defects even in the presence of a NOTCH canonical pathway inhibitor (DAPT), which suggests that JAG1-induced pediatric HBO cell-facilitated bone regeneration

occurs via a NOTCH non-canonical pathway. Therefore, to identify the NOTCH non-canonical pathway targets that are activated by JAG1-bds in HBO cells, we isolated RNA from the JAG1-bds-treated HBO cells, subjected it to RNA sequencing and performed pathway analyses on the data obtained. The data showed that a total of 448 genes were upregulated, and 435 genes were downregulated in JAG1-bds and JAG1-bds + DAPT groups compared to no treatment and thus these genes participate in the non-canonical NOTCH signaling pathway (*Figure 3B*, *Supplementary file 1*). Some of the genes that are upregulated as part of the non-canonical NOTCH pathway are known to be involved in osteoblast commitment (*RUNX2*), matrix remodeling (*MMP3*), and diverse cytokines and chemokines (*CCL5* and *CXCL1*). RUNX2 is commonly known to be expressed by osteoblasts as a sign of their recruitment into the lineage (*Komori, 2010*). MMP3 is also abundantly expressed by osteoblasts and is an important regulator of bone remodeling. MMP3 is an enzyme that is essential in the processing of collagen on bone surface, which in turn is necessary for osteoclast recruitment and bone resorption (*Jehan et al., 2022*). Chemokine CCL5 is abundantly expressed by osteoblasts, and it is also involved in recruitment of osteoblast progenitor cells (*Brylka and Schinke, 2019*). PTH/parathyroid hormone 1 receptor (PTH1R) induces the differentiation of osteoblasts, and it has been shown that CXCL1 serves as an intermediate during this process (*Onan et al., 2009*). GO analysis of the upregulated genes in the non-canonical pathway revealed significant enrichment of GO terms associated with *RUNX2*, cytokine signaling, and cell cycle as described earlier are involved in osteoblastic cell proliferation, recruitment, and differentiation leading to bone remodeling. *RELA* was also upregulated, and it is known to be a radiation-induced pro-survival factor in human osteoblastic cells (*Xiao et al., 2009*). More interestingly, the PIP3 activating AKT signaling pathway was upregulated by JAG1-bds treatment, suggesting that JAG1 can activate NOTCH non-canonical signals via the AKT pathway. Collectively, these data reveal that JAG1 has a profound NOTCH non-canonical effect on HBO cells that stimulates HBO cell-osteoblast commitment and differentiation, and HBO-cell-induced bone formation.

We also obtained lysates from HBO cells treated with JAG1-bds in the presence and absence of DAPT and subjected them to Luminex-based multiplex assays. The Luminex-based assays demonstrated that JAG1 induces the phosphorylation of various signaling targets, including STAT5, AKT, P38, JNK, RELA, and p70 S6K, between 15 and 30 min, as shown in *Figure 4*. As discussed earlier, prior studies emphasize the importance of non-canonical signaling, crosstalk between NOTCH, and other cellular signaling mechanisms (*Caliceti et al., 2014*). These studies show that STAT5 is essential for AKT–p70 S6K activity during lymphocyte proliferation in patients with leukemias and lymphomas (*Lockyer et al., 2007*), and that AKT–mTOR–p70 S6K, ERK, and NF-κB were involved together in proliferation of osteosarcoma cells (*Miwa et al., 2012*). JNK was previously found to phosphorylate p70 S6K to induce osteoblast proliferation and differentiation of MC3T3 cells (*Iijima et al., 2002*; *Svegliati-Baroni et al., 2003*; *Żurek et al., 2019*), and that the P38 pathway has been shown to activate the mTOR–p70 S6K pathway during oxidative stress in mouse embryonic fibroblasts (*Gutiérrez-Uzquiza et al., 2012*). Taken together, our results and the results of others demonstrate that the p70 S6K pathway may be a node at which osteoblast induction occurs in HBO cells, making it a potential major contributor to bone regeneration caused by JAG1-induced HBO cells. To test this hypothesis, we proceeded to confirm that p70 S6K is an essential target of the JAG1-induced NOTCH non-canonical signaling in HBO cells (*Figure 5*). p70 S6K is an enzyme that phosphorylates the S6 ribosomal protein to initiate protein synthesis which supports growth, proliferation, differentiation, and glucose homeostasis of cells (*Ruvinsky et al., 2005*; *Ruvinsky and Meyuhas, 2006*; *Ruvinsky et al., 2009*). We found that JAG1-induced mineralization in HBO cells was predominantly (50.2%) inhibited by S6K-18, an inhibitor of the phosphorylation of p70 S6K, recognizing that other non-canonical signaling pathways are also important in bone regeneration (e.g., p38, AKT). However, inhibition of JAG1-induced mineralization caused by S6K-18 treatment alone was significant compared to that induced by JAG1-bds alone ($p = 0.015$), suggesting that the phosphorylation of p70 S6K is a significant contributor of osteoblast induction in JAG1-stimulated HBO cells. Thus, p70 S6K is an important downstream target of JAG1-NOTCH in JAG1-stimulated HBO cells.

Studying the mechanisms by which JAG1 induces osteoblast commitment and bone formation will enable new treatment avenues that involve the delivery of not only tethered JAG1 but also the JAG1-NOTCH non-canonical signaling intermediates themselves or their activators as powerful treatment options to induce bone regeneration in CF bone loss injuries. High-throughput screening for existent, FDA-approved drugs, compounds, and small molecules can be used to identify pharmacological

activators of p70 S6K, as future directions of this study. Additionally, further investigation of RELA and the other downstream signaling targets identified in our Luminex-based assays are currently in the planning stages. These findings can provide powerful treatment options to induce bone regeneration in CF bone loss injuries and avoid the limitations with currently available therapies.

# Materials and methods

## HBO cell isolation

HBO cell lines were derived from healthy fibulas of seven pediatric subjects under appropriate Institutional Review Board approval. Segments of the fibulas were digested using collagenase A (Roche, 10103578001) used at 0.1 mg/ml treatment for 40 min with replacement of collagenase A midway at 20 min and then digested with 0.2 mg/ml of collagenase A for 60 min at 37°C with intermittent shaking. The segments of bone were maintained in Dulbecco's modified Eagle medium (DMEM) + Primocin Antimicrobial agent for primary cells (Invivogen, ant-pm-1) + 10% fetal bovine serum (FBS) + 50 µM ascorbic acid (Sigma, 49752) + 10 nM dexamethasone. Media changes were performed every 5 days. Osteoblast-like cells began to grow out of the bone segments by day 10. These cells can be passaged and frozen for storage. HBO cell lines used for experiments were selected based on the ability of the primary cell line to proliferate and mineralize in culture. On addition of Osteogenic media [DMEM + Primocin + 10% FBS + 50 µM ascorbic acid + 10 nM dexamethasone (Sigma, D1756) + 10 mM beta-glycerophosphate disodium (Sigma, G9422)], the cells form mineralized nodules indicating their osteogenic ability as seen in *Figure 1—figure supplement 1*.

## JAG1 immobilization

As per the manufacturer's recommendations, 50 µl (1.5 mg) of Dynabeads Protein G (Invitrogen 10004D) were transferred to a tube, where the beads were separated from the solution using a magnetic tube rack (Bio-Rad, 1614916) and washed once using 200 µl phosphate-buffered saline (PBS) with 0.1% Tween-20 (Fisher, BP337-500). The wash buffer was separated from the beads-Fc complex using the magnetic rack. Recombinant JAG1-Fc (5, 5.7, 10, or 20 µM) (Creative Biomart, JAG1-3138H) or control IgG-Fc fragment (5 or 5.7 µM) (Abcam, ab90285) were diluted in 200 µl PBS with 0.1% Tween-20 and then added to the Dynabeads. The beads plus proteins were incubated at 4°C with rotation for 16 hr. Thereafter, the tubes were placed back on the magnetic rack and the supernatant was removed. The bead–JAG1/Fc complex was resuspended in 200 µl PBS with 0.1% Tween-20 to wash by gentle pipetting. The wash buffer was also separated from the beads–Fc complex using the magnetic rack, and the final suspension of the beads in hydrogels was used as treatment.

## Mineralization assay

HBO cells were seeded at 30,000 cells per well in a 12-well plate, treated ($n$ = 3) with and cultured for 21 days in osteogenic media (DMEM + Primocin + 10% FBS + 50 µM ascorbic acid + 10 nM dexamethasone + 10 mM beta-glycerophosphate) with half feeds every 5 days. Treatments included growth media (DMEM + Primocin + 10% FBS), osteogenic media, Fc-Dynabeads (5.7 µM) in osteogenic media or JAG1-Dynabeads (5.7 µM) in osteogenic media. Inhibitors N-[N-(3,5-difluorophenacetyl)-L-alanyl]-S-phenylglycine t-butyl ester (DAPT; Sigma, D5942) (15 µM) and 5-(1,1-dimethylethyl)–2-[[(1H-indazol-5-ylamino)carbonyl]amino]-3-thiophenecarboxylic acid (S6K-18 – inhibitor of p70 ribosomal S6 kinase 1; Selleck Chemicals, S0385) (50 µM) were also added to JAG1-Dynabeads conditions for experiments shown in *Figure 5* and *Figure 5—figure supplements 1 and 2*. For PCR, cells were collected on days 7–9, 14, and 21 (see Methods 'RNA extraction and qRT-PCR'). For Alizarin Red staining: On day 21, the cells were fixed using 50% ethanol for 15 min at 4°C. The fixed cells were then stained with Alizarin Red S dye (LabChem, LC106002) to detect mineralization. The dye was extracted using a 10% acetic acid solution in water and quantified by measuring the absorbance at 420 nm using a spectrophotometer.

## Hydrogel preparation

We prepared poly (ethylene glycol) (PEG)-based synthetic hydrogels incorporating cell adhesive peptides in two steps. First, maleimide end-functionalized 20 kDa four-arm PEG macromer (PEG-4MAL, with >95% end-group substitution, Laysan Bio, 4ARM-PEG-MAL-20K) was reacted with a

thiol-containing adhesive peptide GRGDSPC (RGD, Genscript, RP20283) in PBS with 20 mM 4-(2-hy droxyethyl)-1-piperazineethanesulfonic acid (HEPES) at pH 7.4 for 1 hr. Then, the RGD-functionalized PEG-4MAL macromers were cross-linked in the presence of HBO cells and JAG1-Dynabeads into a hydrogel by addition of the dithiol protease-cleavable peptide cross-linker GPQ-W (GCRDGPQGIW-GQDRCG) (New England Peptides, Inc (NEP) Custom synthesized) (*Kamalakar et al., 2019*; *Phelps et al., 2012*). The final gel formulation consisted of 4.0% wt/vol polymer and 1.0 mM RGD.

## In vivo experiments

We performed all in vivo experiments using procedural guidelines with appropriate approvals from the Institutional Animal Care and Use Committee of Emory University (#PROTO201700263). Six- to eight-week-old male and female NOD.Cg-Prkdc^scid/J mice (The Jackson Laboratory, 001303) were used. As shown in *Figure 2—figure supplement 1*, the surgery site was disinfected and then incisions were made using sterile surgical equipment to expose the parietal bones of the mice. Thereafter, 4 mm defects were created in the parietal bones using a variable speed drill (Aseptico (MicroNX), MAX-88ESP, CL1791023) and sterile circular knives. *JAG1 delivery*: PEG-4MAL hydrogels (20 µl) loaded with JAG1-Dynabeads without or with 100,000 HBO cells (*n* = 13–15 for all treatment groups) were placed within the defects created in parietal bones in the NOD-SCID mouse skulls as the first dose. A second dose of the hydrogels encapsulating HBO cells and Dynabead-bound JAG1 were administered as transcutaneous injections during week 4 to continue the bone regenerative action of JAG1. Skulls were then harvested at week 8, fixed using 10% neutral buffered formalin (VWR, 89370-094) and imaged with µCT. HBO primary cell lines 2, 6, and 7 from separate individuals were selected for these experiments based on similar growth and passage characteristics. Inclusion criteria: Survival to end of the experiment. Exclusion criteria: Illness that required euthanasia before the end of the experiment or death.

## Micro computed tomography

µCT analyses were conducted according to current guidelines for the assessment of BV within the defects created in mouse calvaria (*Bouxsein et al., 2010*). Briefly, formalin-fixed skulls were posi-tioned in the µCT tubes with the nose facing the bottom of the tube and imaged in a µCT 40 (Scanco Medical AG, Bassersdorf, Switzerland) using a 36-µm isotropic voxel size in all dimensions. Thereafter, using a consistent and pre-determined threshold of 55 kVp, 145 µA, 8 W and 200 ms integration time for all measurements, three-dimensional reconstructions were created by stacking the regions of interest from ~600 two-dimensional slices consisting of the entire skull and then applying a gray-scale threshold of 150 and Gaussian noise filter ($\sigma$ = 0.8, support = 1.0), a coronal reformatting was done. Thereafter, a circular region of interest encompassing the defect was selected for analysis consisting of transverse CT slices encompassing the entire defect and, new BV was calculated.

## Histology

### Masson trichrome staining

Formalin-fixed skulls used for µCT measurements were decalcified in Cal-ex (Fisher, C5510-1D), embedded in paraffin, and sectioned on a microtome to obtain 5 µm sections which were then stained using a Masson Trichrome staining kit (Sigma Aldrich, HT15) according to the manufacturer's protocol. The sections were first washed with PBS, three times, for 5 min each, then the slides were submerged in Bouin's solution for 15 min, and then washed under running water for 5 min. Thereafter, the sections were incubated in Weigert's working hematoxylin solution for 10 min before being washed three times under running water for 5 min each and then transferred to distilled water. The sections were then stained with biebrich scarlet acid fuchsin for 5 min and washed with distilled water three times before being submerged in a phosphotungstic/phosphomolybdic solution for 10 min, and subse-quently placed in an analine blue solution. Lastly, they were washed with distilled water three times and moved to a solution of 1% acetic acid for 1 min followed by a submersion in two changes of xylene and mounted with a coverslip. The sections were then imaged using brightfield microscopy.

### Immunohistochemical analysis

Deparaffinization of FFPE sections was first performed by incubating them for 45 min at 60°C followed by xylene (two times, 5 min each) and then sections were rehydrated using various grades of alcohol in

a decreasing concentration (100%, 90%, and 75%). The sections were washed with 1× PBS for 10 min, permeabilized with 1.0% Triton X-100 (in 1× PBS) for 10 min at room temperature and rinsed with 1× PBS (three times, 5 min each). The sections were covered with 20 µg/ml of Proteinase K in 1× PBS for 20 min at 37°C in a humidified chamber. After a 1× PBS rinse (two times, 5 min each), the sections were blocked with 1% bovine serum albumin (Fraction V, BP1600-100) for 30 min at room temperature. The sections were then washed with 1× PBS (two times, 5 min each) and incubated with the primary antibody (COL1A1, Cell Signaling #72026S, RRID:AB_2904565) at 1:50 dilution for 24 hr at 4°C. Recombinant rabbit IgG monoclonal antibody was used as a negative isotype control. Sections were washed with 1× PBS (three times, 5 min each), covered with two drops of SignalStain Boost Detection Reagent (HRP, Rabbit #8114), and incubated in a humidified chamber for 30 min at room temperature. After 1× PBS wash (three times, 5 min), SignalStain DAB Substrate (Kit #8059) was applied for 5 min. The slides were immersed in de-ionized water (dH$_2$O) for 5 min, and then counterstained with hematoxylin [Vintage Hematoxylin (SL 100)] for 5 s. After a 5-min wash with dH$_2$O, the sections were dehydrated with alcohol in increasing concentration (95% and 100%) followed by two incubations in xylene for 10 s each. The slides were mounted and coverslipped using Permount Mounting Medium (#17986-05). Slides were scanned with the Olympus Nanozoomer whole-slide scanner at 20×.

## RNA extraction, qRT-PCR, and RNA-sequencing

### RNA extraction and quantitative reverse transcription polymerase chain reaction (qRT-PCR)

HBO cells were grown in biological triplicates in growth media alone, osteogenic media alone or with Fc-Dynabeads (5.7 µM) or JAG1-Dynabeads (5.7 µM). Cells were collected at 7–9, 14, and 21 days. RNA was extracted using TRIzol (Invitrogen, 15596026) by adding the TRIzol (400 µl to 1 ml) directly to plates of adherent cells and collecting by pipette. Samples were frozen at −80°C and then processed in batches. Samples were allowed to thaw, and the appropriate volume of chloroform was added (200 µl per ml of TRIzol), samples were vortexed for 1 min, allowed to rest for 5 min and then centrifuged at 12,000 × $g$, 15 min, 6°C. After centrifugation, the aqueous phase was removed to a separate tube and cold isopropanol was added (600 µl isopropanol per ml TRIzol). Samples were mixed well and kept at −20°C for 10 min followed by centrifugation at 12,000 × $g$, 10 min, 6°C. Supernatant was removed and cold 75% ethanol/25% RNase free water was added (500 µl per ml Trizol). Samples were vortexed and centrifuged at 12,000 × $g$, 5 min, 6°C. Supernatant was removed and samples were allowed to air dry for 10 min. RNA was resuspended in 20–30 µl of RNase free water, warmed at 55°C for 10 min, and then quantified using a NanoDrop One spectrophotometer. The High Capacity cDNA Reverse Transcription Kit (Ref 4368814) from Applied Biosystems was used to produce cDNA following the manufacturer's instructions. Primers were acquired from Integrated DNA Technologies (see *Supplementary file 2* for sequences) and qRT-PCR was done in duplicate using Bio-Rad iQ SYBR Green Supermix (Bio-Rad, 1708882) following the manufacturer's instructions. Data were normalized to growth media condition with glyceraldehyde-3-phosphate dehydrogenase as the reference gene. Analysis of qRT-PCR was accomplished by using double delta Ct analysis in Excel and data were plotted in GraphPad Prism 10.

### RNA sequencing

RNA was isolated using the QIAGEN RNeasy kit (QIAGEN, 74106) according to the manufacturer's protocols. The samples were submitted to the Molecular Evolution core at the Georgia Institute of Technology for sequencing. Quality control (QC) was performed using an Agilent Bioanalyzer 2100 to determine the RNA integrity number (RIN) of the samples. mRNA was enriched using the New England Biolab's (NEB) NEBNext Poly(A) mRNA isolation module for samples with RINs greater than 7 and libraries were prepared using the NEBNext Ultra II directional RNA library preparation kit (NEB, E7760). QC was then performed on these libraries using an Agilent Bioanalyzer 2100 and the libraries were quantified using fluorometric methods. Paired-end 150 base pairs (PE150) sequencing was performed on the Illumina NovaSeq 6000 instrument to obtain a sequencing depth of 30 million reads per sample. The transcripts obtained were aligned using the hg38 genome reference database along with elimination of duplicate reads, using the DNAStar Lasergene 17.3 application. The RNA levels were calculated in reads per kilobase per million mapped reads (RPKM). Genes expressed at >1.5 RPKM were retained for further analyses.

## Transcriptomic analysis methods

### Differential gene expression and enrichment analysis

DEGs were determined using DESeq2 (v1.38.3) available in R Bioconductor (*Love et al., 2014*). Transcripts with an FDR-adjusted p-value <0.05 were considered significant for this analysis. Transcript counts were normalized using the median-of-ratios method used by DESeq2 prior to differential expression analysis. Results were visualized with Venn diagrams using govenn (v0.1.10), volcano plots in R using ggplot2 (v3.4.1) (H. Wickham. Ggplot2: Elegant Graphics for Data Analysis. Springer-Verlag New York, 2016), and heatmaps in R using Heatmap3 (v1.1.9) (*Zhao et al., 2014*).

### Transcript functional annotation

To provide additional functional annotation of the DEGs, a web-scraper was built to parse the National Center for Biotechnology Information Gene Database (*Sayers et al., 2021*) entry for each transcript and identify relevant terms. Transcript names were connected to the NCBI Entrez ID with the Genome wide annotation for Human package, org.Hs.eg.db (v3.15.0), available through Bioconductor in R Carlson M (2019). *Org.Hs.eg.db: Genome wide annotation for Human.* R package version 3.15.0. This tool utilized the HTML parser in the BeautifulSoup4 (v4.12.2) Python package to extract the gene summary information from each gene entry in the database. The script then identified the keywords in the summary.

### Functional over-representation analysis

Over-represented GO terms were identified using PANTHER (v17.0) (*Thomas et al., 2022*; *Mi et al., 2019*). DEGs involved in the non-canonical signaling pathway were identified from Venn diagrams and separated as up- or downregulated relative to control. Each list was uploaded to the PANTHER online tool for over-representation testing. FDR-adjusted Fischer's test $p < 0.05$ was considered over-represented in this analysis. The complete GO Biological Process term list was evaluated and the complete list of identified transcripts from RNAseq was set as the background.

### Luminex-based Multiplex assay

To detect phosphor-signaling targets, serum-starved HBO cell lines ($n = 3$) from different patients were treated ($n = 3$ per cell line) with Fc-Dynabeads (5.7 µM), unbound BMP2 (100 nM), and JAG1-dynabeads (5.7 µM) with or without DAPT (15 µM) as a time course stimulation for 5, 10, 15, and 30 min. Whole-cell protein (2 µg) lysates were subjected to a Millipore Luminex-based Multiplex assay to measure signaling targets using the Milliplex multiple pathway cell signaling magnetic bead 9-Plex kit (Millipore Sigma, 48-681MAG) according to the manufacturer's protocol.

### Statistics

Data were analyzed by analysis of variance with Tukey's post hoc test unless otherwise noted using GraphPad Prism 10. All data are presented as mean ± standard deviation. $p < 0.05$ between groups was considered significant and are reported as such. MATLAB coding was used to create heatmaps and to generate *z*-score values associated with color intensities seen on the heatmap.

## Acknowledgements

Study design: AK, AIT, SA, AJG, LBW, HD, MHR, TC, SC, and SLG. Study conduct: AK, SK, and MHR. Data collection: AK, BT, SK, MHR, and IM. Data analysis: AK, BT, SK, MHR, AIT, and IM. Data interpretation: All authors. Drafting manuscript: AK and SLG. Revising manuscript content: All authors. Approving the final version of manuscript: All authors. We extend our gratitude to Adrianna Westbrook and Katie Liu from the Pediatric Biostatistics Core, Emory University, for their invaluable advice on how best to perform the statistics on the data in this manuscript. AK and SLG take responsibility for the integrity of the data analysis. Research reported in this publication is supported by the National Institutes of Health, National Institute of Dental and Craniofacial Research (NIDCR) under award number R01DE031271 and National Institutes of Health, National Institute of Arthritis and Musculoskeletal and Skin Diseases of the National Institutes of Health under Award Numbers DE026762, R01AR062920, and R01AR062368, respectively. The authors declare no conflict of interest.

## Additional information

### Funding

| Funder | Grant reference number | Author |
|---|---|---|
| National Institute of Dental and Craniofacial Research | 1R01DE031271-01-A11 | Andres Garcia Levi Wood Steven L Goudy |
| National Institute of Arthritis and Musculoskeletal and Skin Diseases | DE026762 | Steven L Goudy |
| National Institute of Arthritis and Musculoskeletal and Skin Diseases | R01AR062920 | Andres Garcia |
| National Institute of Arthritis and Musculoskeletal and Skin Diseases | R01AR062368 | Andres Garcia |

The funders had no role in study design, data collection, and interpretation, or the decision to submit the work for publication.

### Author contributions

Archana Kamalakar, Formal analysis, Investigation, Writing – review and editing, Conceptualization, Data curation, Methodology, Writing – original draft, Project administration; Brendan Tobin, Writing – review and editing, Project administration; Sundus Kaimari, M Hope Robinson, Timothy Cha, Writing – review and editing, Conceptualization, Project administration; Afra I Toma, Conceptualization, Methodology; Samir Chihab, Irica Moriarity, Surabhi Gautam, Pallavi Bhattaram, Conceptualization; Shelly Abramowicz, Resources; Hicham Drissi, Andres Garcia, Levi Wood, Resources, Supervision, Project administration; Steven L Goudy, Formal analysis, Resources, Supervision, Funding acquisition, Methodology, Project administration

### Author ORCIDs

M Hope Robinson ⓘ https://orcid.org/0000-0002-4255-1308
Hicham Drissi ⓘ https://orcid.org/0000-0002-3322-281X
Steven L Goudy ⓘ https://orcid.org/0000-0002-8255-3965

### Ethics

We performed all in vivo experiments using procedural guidelines with appropriate approvals from the Institutional Animal Care and Use Committee of Emory University, #PROTO201700263.

Reviewer #1 (Public review): https://doi.org/10.7554/eLife.92925.3.sa1
Reviewer #2 (Public review): https://doi.org/10.7554/eLife.92925.3.sa2
Author response https://doi.org/10.7554/eLife.92925.3.sa3

## Additional files

### Supplementary files

• Supplementary file 1. List in excel sheet attached shows differentially expressed genes (DEGs) in JAG1 and JAG1 + DAPT groups compared to no treatment. Analysis revealed a total of 448 upregulated genes and 435 downregulated genes in the non-canonical pathway.

• Supplementary file 2. Primer sequences used for qRT-PCR.

• MDAR checklist

1231Data availability
Bulk RNAseq data generated through this work have been deposited in the Gene Expression Omnibus (GEO) database with accession number GSE274096.

The following dataset was generated:

| Author(s) | Year | Dataset title | Dataset URL | Database and Identifier |
|---|---|---|---|---|
| Kamalakar A, Tobin B, Kaimari S, Robinson MH, Toma AI, Cha T, Chihab S, Moriarity I, Gautam S, Bhattaram P, Abramowicz S, Drissi H, García AJ, Wood LB, Goudy SL | 2024 | Delivery of a Jagged1-PEG-MAL hydrogel with pediatric human bone cells regenerates critically sized craniofacial bone defects | https://www.ncbi.nlm.nih.gov/geo/query/acc.cgi?acc=GSE274096 | NCBI Gene Expression Omnibus, GSE274096 |

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
