## [Editor Report · eLife assessment]

Therapeutic treatments for congenital and acquired craniofacial (CF) bone abnormalities are not well developed. This study provides **convincing** evidence for an innovative regenerative treatment for pediatric craniofacial bone loss using Jagged1-PEG-MAL hydrogel with pediatric human bone cells. The report is a **valuable** advance in this field.

---

## [Referee Report · Reviewer #1 (Public review)]

Summary:

In this manuscript, the authors conducted an important study that explored an innovative regenerative treatment for pediatric craniofacial bone loss, with a particular focus on investigating the impacts of JAGGED1 (JAG1) signaling.

Strengths:

Building on their prior research involving the effect of JAG1 on murine cranial neural crest cells, the authors demonstrated successful bone regeneration in an in vivo murine bone loss model with a critically-sized cranial defect, where they delivered JAG1 with pediatric human bone-derived osteoblast-like cells in the hydrogel. Additionally, their findings unveiled a crucial mechanism wherein JAG1 induces pediatric osteoblast commitment and bone regeneration through the phosphorylation of p70 S6K. This discovery offers a promising avenue for potential treatment, involving targeted delivery of JAG1 and activation of downstream p70 s6K, for pediatric craniofacial bone loss. Overall, the experimental design is appropriate, and the results are clearly presented.

---

## [Referee Report · Reviewer #2 (Public review)]

The current manuscript undoubtedly demonstrates that JAG1 can induced osteogenesis via non-canonical signaling. In fact, using the mouse-calvarial critical defect model, the authors have clearly shown the anabolic regenerative effect of JAG1 in via non-canonical pathways. Exploring the molecular mechanisms, the authors have shown that non-canonically JAG1 is regulating multiple pathways including STAT5, AKT, P38, JNK, NF-ĸB, and p70 S6K, which together possibly culminate to the activation of p70 S6K. In summary these findings have significant implications in designing new approaches for bone regenerative research.

---

## [Author Response]

The following is the authors’ response to the original reviews.

**Reviewer #1 (Recommendations For The Authors):**
Major comments:(1) Regarding the cell studies of human pediatric bone-derived osteoblast-like cells (HBO), the authors should provide a rationale for their selection of specific cell lines (15,16, 17, 19, 20, 23, 24) in this study. As for animal studies, could the authors clarify which cell lines were utilized in the murine in vivo experiments?

We appreciate the opportunity to address this. To reduce confusion, we have numbered the patient primary cell lines used in these studies sequentially from 1 – 7. Additionally, we have added “HBO cell lines used for experiments were selected based on the ability of the primary cell line to proliferate and mineralize in culture” to the Methods section.

In vivo experiments: “HBO cell lines 2, 6 and 7 from separate individuals were selected for these experiments based on similar growth and passage characteristics.” This statement is included in the Methods section.

(2) In this study, the authors performed the murine in vivo experiments using both male and female mice. Could the author clarify if any difference was observed between male and female mice in the findings? This information would contribute to a more comprehensive understanding of the study.

We agree and have added the following to the Results section: “There was no sex-based difference in regenerated bone volume.”

(3) Although the histological results showed an elevated collagen expression in mice treated with BMP2, JAG1, and JAG1 + DAPT compared to those treated with the cells alone, the differences among groups were subtle. The authors should consider the immunohistochemical (IHC) staining for collagen 1 on the samples, allowing for a quantitative assessment of collagen 1 expression.

Thank you for this comment. The differences between BMP2, JAG1, and JAG1 + DAPT are indeed subtle. We have added Supplementary Figure 5, showing collagen staining of sections from the same FFPE blocks that were sectioned and stained with Masson Trichrome in Figure 2C.

Minor Comments:(4) Please specify which cell lines are represented in the staining results shown in Fig.1A and Fig. 5A, respectively.

In Fig 1A the representative images are of HBO2. Fig 5A representative images are of HBO7. We have added this information to the figure legends for these figures.

(5) There appears to be a discrepancy in the specified size of the critical defect. The manuscript states that the size is 4mm, while Supplemental Figure 3 indicates 3.5mm.

Thank you for this catch! Yes, it should be 4mm. This has been corrected in Supplementary Figure 3.

(6) The scale bar for Figure 2 C is missing.

Scale bars have been added which also gave us an opportunity to brighten the images equally, allowing for better distinction between the different colors of the Masson Trichrome staining.

(7) In the methodological section 2.5 for JAG1 delivery, it would be helpful if the authors could review the initial dosage of JAG1 delivery to confirm if HBO cells were included or not, given that the MicroCT results indicate that all groups incorporated HBO cells.

We appreciate this suggestion. In response to another question, we have added Supplementary Figure 4 which includes an “Empty Defect” condition with no HBO cells, making the original method statement accurate.

**Reviewer #2 (Recommendations For The Authors):**
In the current study, using in vitro and in vivo models the authors clearly show that JAG1 can enhance osteogenesis and thus can be helpful in designing new therapeutic approaches in the field of bone regenerative research. The in vivo mouse CF model is very convincing and shows that JAG1 promotes osteogenesis via non-canonical signaling. Mechanistically it seems that JAG1 activates STAT5, AKT, P38, JNK, NF-ĸB, and p70 S6K. However, additional evidence is needed to convincingly conclude that all the non-canonical pathways activated via JAG1 converge at p70 S6K activation. The following concerns need to be addressed.(1) In Fig 1A: Even though the Jag1-Fc shows a very significant increase in HBO mineralization, there are no significant increases in cells in osteogenic media when compared to control growth media. Even though the different conditions were subjected to RNAseq analysis in the later figures, qPCR analysis of some osteogenic genes in Figure 1 might be helpful.

We appreciate the opportunity to explore this question further. We conducted mineralization experiments in triplicate and performed qRT-PCR, assessing for gene expression of 5 osteogenic genes: ALPL, BGLAP (osteocalcin), COL1A1, RUNX2, and SP7. Results are shown in Figure 1C and this text was added to Results: “Additionally, PCR analysis of HBO1 cells from a repeat experiment collected at days 7, 14, and 21 showed significantly increased expression of osteogenic genes with JAG1-bds stimulation (Figure 1C). ALPL was significantly expressed at Day 7, with a 3.5-fold increase (p=0.0004) compared to HBO1 cells grown in growth media. In contrast, significant expression levels of COL1A1 and BGLAP were observed at 14 days, with a 5.1-fold increase (p=0.0021) of COL1A1 and a 12.3-fold increase (0.0002) of BGLAP when compared to growth media conditions. Interestingly, while some mineralization is observed in the osteogenic media and Fc-bds

(Figure 1A) conditions, there were no significant increases in osteogenic gene expression (Figure 1C). Expression of RUNX2 and SP7 was not significantly altered across all conditions and time points (not shown).”

(2) In Fig 2: even though not needed in respect to the hypothesis, was there any Control group without any cells or JAG1 beads? What were the changes in between that group and cells cells-only group?

We have not observed differences between the “Empty Defect” group and the “Cells alone” group.

We have addressed the reviewer’s comments by adding this comparison in Supplementary Figure 4.

(3) Transcriptional profiling and ELISA (Fig 3 and 4) show upregulation of NF-ĸB signaling in response to JAG1. In the discussion, the authors have referenced a previous study showing NF-ĸB as prosurvival in human OB cells. However, based on many published reports, NF-ĸB activation has been shown to inhibit OB function. Does JAG1 regulate HBO cell survival via NF-ĸB activation?Experimenting using NF-ĸB inhibitor can be helpful to show that JAG1 mediates NF-ĸB activation is anabolic in this experimental setup.

We thank the reviewer for this excellent suggestion. We are eager to explore this new direction for our research in a subsequent study. We have added this to our future directions.

(4) Fig 5:(A) Condition showing JAG1+ DAPT is needed to compare between JAG1 canonical and noncanonical signaling.

Thank you for pointing this out. We have added Supplementary Figure 6, which includes a dose response experiment for JAG1 + DAPT.

(B) S6K18 alone seems to be increasing OB mineralization. Is that statistically significant?

No, and we have added the statistical analysis for S6K-18 to Figure 5B.

(C) Fc alone condition seems to have a very significant increase in OB mineralization. Does Fc alone upregulate OB function?

We do see some upregulation of mineralization with Fc in vitro, which we also observed in our previous studies with mouse neural crest cells, but we have not found it to be osteogenic in vivo. We have added a statement to this effect, with references. Additionally, osteogenic gene expression was not upregulated in our in vitro mineralization experiments with Fc. See Revised Figure 1.

(D) Although overall quantification shows that S6K18 partially inhibits HBO mineralization, the representative images do not represent the quantification. Transcriptional analysis (qPCR) is required to validate these findings.

We performed qRT-PCR on cells from a repeat mineralization assay, collecting cells at 9, 14, and 21 days. We have added the following to the Results:” While inhibition of NOTCH and p70 S6K decreased mineralization in our mineralization assay, there are no statistically significant changes in gene expression for ALPL, COL1A1, or BGLAP (Supplementary Figure 7). These results suggest that the HBO cells phenotypes are maturing into osteocytes and that inhibiting p70 S6K hinders the cellular ability to mineralize but not the cell phenotype progression.”

(5) Finally, to convincingly conclude the data from Fig 5, the mouse CF model can be helpful to support the authors' claim that JAG1 acts via p70 S6K.

Thank you for this feedback. We have modified our conclusions to reflect that p70 S6K is one of the non-canonical pathways that JAG1 may be activating in bone regeneration.

Thank you very much for your consideration of our revised manuscript.